# OpenReview forum: "Who’s in Charge? Disempowerment Patterns in Real-World LLM Usage"
_ICML.cc/2026/Conference — ICML 2026 regular_

### Official Review · Reviewer_ExsS · 2026-03-11

**Soundness:** 3
**Presentation:** 3
**Significance:** 4
**Originality:** 4
**Overall Recommendation:** 5
**Confidence:** 3

**Summary:**

This paper studies situational disempowerment potential in real-world AI assistant interactions. It proposes a framework with three primitives (reality distortion potential, value judgment distortion potential, and action distortion potential) along with four amplifying factors, and applies a privacy-preserving analysis pipeline to 1.5 million Claude.ai conversations. The paper reports that severe disempowerment potential is rare overall but more common in personal domains, and that these interactions tend to receive higher user approval ratings.

**Compliance With Llm Reviewing Policy:**

Affirmed.

**Final Justification:**

The rebuttal has fully addressed my original questions. I maintain my original rating (5 Accept).

**Key Questions For Authors:**

1. How does the proposed framework relate to existing theory on autonomy, dependency, social influence, or related constructs?
2. What is the rationale for the chosen amplifying factors, and how are potentially overlapping factors empirically distinguished?
3. Can the authors discuss dataset representativeness and possible temporal/geographic/cultural biases more explicitly?
4. How should readers interpret linguistic cues or examples that may have culturally specific meanings (e.g., sensei, master, mentor)? Clarifying this would help temper overgeneralization from examples and improve interpretability.

**Limitations:**

Yes

**Strengths And Weaknesses:**

This paper addresses an important and timely societal dimension of LLM use, and the large-scale privacy-preserving analysis of real-world conversations is a notable strength. I also found the decomposition into three disempowerment-potential primitives to be a useful organizing framework. My main concerns are about conceptual grounding and interpretive scope. First, Section 2 would benefit from stronger positioning of the disempowerment framework in adjacent literature, especially from social science/HCI. Second, the amplifying factors in Section 3.1 would benefit from stronger theoretical support and clearer differentiation, since some categories (e.g., attachment and reliance/dependency) appear potentially overlapping. Third, the paper should more explicitly discuss what population the dataset represents and how temporal, geographic, and cultural factors may limit or bias the findings. Overall, I appreciate the authors for examining this important issue, but I think the framing and limitations discussion would benefit from further strengthening.

---

> ### Author Rebuttal · Authors · 2026-03-30
>
> Thank you for the detailed feedback.  We agree with most of it.  To address your key questions in turn:
>
> 1) “How does the proposed framework relate to existing theory on autonomy, dependency, social influence, or related constructs?”
>
> The main source of inspiration for this analysis, which we cited, was:
>
>  - Human Autonomy at Risk? An Analysis of the Challenges from AI by C. Prunkl
>     https://link.springer.com/article/10.1007/s11023-024-09665-1
>
> However, the framework was derived mainly from our own first-principles thinking.
>
> Since our initial submission, we’ve identified other related work, such as:
> 1. Laitinen & Pessi, "AI Systems and Respect for Human Autonomy" (Frontiers in AI, 2021)
> https://www.frontiersin.org/journals/artificial-intelligence/articles/10.3389/frai.2021.705164/full
> 2. Susser, Roessler & Nissenbaum, "Online Manipulation: Hidden Influences in a Digital World" (Georgetown Law Technology Review, 2019)
> https://papers.ssrn.com/sol3/papers.cfm?abstract_id=3306006
> 3. Carmona-Díaz, Jiménez-Leal, Vold et al., "Autonomy by Design: Preserving Human Autonomy in AI Decision-Support" (Philosophy & Technology, 2025)
> https://link.springer.com/article/10.1007/s13347-025-00932-2
> 4. Calvo, Peters, Vold & Ryan, "Supporting Human Autonomy in AI Systems: A Framework for Ethical Enquiry" (Springer, 2020)
> https://link.springer.com/chapter/10.1007/978-3-030-50585-1_2
> 5. Christman, "Autonomy in Moral and Political Philosophy" (Stanford Encyclopedia of Philosophy)
> https://plato.stanford.edu/entries/autonomy-moral/
>
> We’re all ears for other relevant work we might have missed.
>
>
> 2) “What is the rationale for the chosen amplifying factors, and how are potentially overlapping factors empirically distinguished?”
>
> The primitives were derived from ad-hoc exploratory analysis of thumbs data.  Conversations can be coded as containing more than one factor if there is overlap.
>
> 3) Can the authors discuss dataset representativeness and possible temporal/geographic/cultural biases more explicitly?
>
> These are important considerations. We note in Section 4.3 that our analysis is limited to Claude.ai traffic, which limits the generalizability of our findings. Also, the composition of Claude’s user base changed considerably during the year included in the feedback data analysis. Recent work from Anthropic’s Economic Index (Appel et al., 2025) highlights the uneven distribution of Claude adoption around the world, and cultural differences may cause systematic over- or under-representation of disempowerment-related metrics. For example, norms around deference to authority vary across cultures, which could affect the prevalence of authority projection. Norms around disclosing vulnerability or distress may also vary by cultural context. We will update the paper to note these potential biases more explicitly.
>
> 4) How should readers interpret linguistic cues or examples that may have culturally specific meanings (e.g., sensei, master, mentor)?
>
> The examples presented in the paper are drawn from cluster descriptions written in English, generated by a language model that reviewed summaries of the underlying transcripts. As a result, terms like "sensei," "master," or "mentor" reflect the summarizing model's characterization of observed dynamics (potentially in languages other than English), and not necessarily the user's original intent. Any transcript-level analysis pipeline, without direct knowledge of the user's cultural context, has limited ability to distinguish these cases. We will add a note to this effect in the paper.

---

> > ### Author Rebuttal · Reviewer_ExsS · 2026-04-03
> >
> > I appreciate the authors' detailed clarifications and the additional references provided. My original questions have been fully resolved.

---

### Official Review · Reviewer_gN8F · 2026-03-18

**Soundness:** 2
**Presentation:** 3
**Significance:** 3
**Originality:** 3
**Overall Recommendation:** 4
**Confidence:** 4

**Summary:**

The authors introduce a taxonomy and measurement framework for "AI disempowerment" - the reduction of human autonomy through interactions with AI assistants. The authors define three dimensions of disempowerment: reality distortion, value judgment distortion, and action distortion. Using Anthropic's privacy-preserving Clio framework, they classify large-scale Claude conversations, finding that while severe disempowerment is rare in absolute terms, it is non-negligible within relevant conversation subsets. A notable empirical finding is that disempowering conversations receive higher user approval (thumbs-up) rates than non-disempowering ones, suggesting a tension between user satisfaction and user welfare. The paper also examines historical trends, finding gradual increases in disempowerment rates over a 14-month window.

**Compliance With Llm Reviewing Policy:**

Affirmed.

**Final Justification:**

The rebuttal clarifications were helpful - my assessment remains positive.

**Key Questions For Authors:**

1. I am concerned that the definition of "disempowerment" is somewhat narrow, focusing primarily on the delegation of moral values and actions - especially in the second and third definitions of "disempowerment potential." For example, value judgment distortion potential is defined as cases where users delegate "moral judgments" and their understanding of values to AI, and action distortion potential as cases where users' "value-laden actions" are largely delegated to AI. However, disempowerment may also arise in non-moral contexts. E.g., reduced critical thinking in situations where users engage in conversation about factual questions (https://arxiv.org/abs/2508.09297). I wonder if the authors intentionally restrict their definition of disempowerment to moral and value-related contexts, and if so, what the rationale is.

2. As the authors' definitions of disempowerment focus on moral and value-related conversations, why don't the authors report the proportion of disempowerment "within" conversations about beliefs, moral decisions, and moral actions? Out of the 1.5 million conversations the authors have access to, only 7.35% seem to be moral/value-relevant conversations, according to the screening results (Appendix J.4, K.3). What's the proportion of disempowerment within these 7.35% conversations? In Fig. 1, the proportion of reality distortion (mild or above) is around 1%, but when focusing on post-screened, moral/value-relevant conversations, I guess it would be around 14% (1 / 7.35). It would be striking if users face reality distortion 1 out of 7–8 times when asking moral/value-related questions.

3. According to the authors' definition of value-distortion disempowerment in Appendix K.2.2, they focus on cases where users already possess prior values that AI overrides or distorts. How does the framework handle cases where users have no pre-existing values on a topic and use AI for "value formation" rather than value distortion? For instance, a young user with no established moral stance on a dilemma who adopts the AI's framing wholesale. Would the current measurement approach categorize them as disempowerment?

4. The corpus spans multiple Claude models (Sonnet 4.5: 62%, Haiku 4.5: 22%, Opus 4.5: 13%). Why is there no model-level breakdown of disempowerment rates or user approval rates? Do more capable models produce more persuasive or coherent responses that increase disempowerment potential? This could partially explain why increases in disempowerment correlate temporally with the releases of Sonnet 4 and Opus 4 (Appendix L), and would be practically useful.

5. Figure 6 compares thumbs-up rates for disempowering vs. non-disempowering conversations as an unconditional comparison, with no regression controls. Without controlling for conversation length, domain, model version, or user-level characteristics - all of which affect user experience - how robust is the claim that users approve of disempowering interactions? Could the authors report results from a linear regression controlling for at least conversation length, domain, and model version? For example, Y = b₁X₁ + b₂X₂ + b₃X₃ + …, where Y is thumbs-up, X₁ is disempowerment, and X₂ onward are relevant controls.

6. Classifier validation in Appendix I relies on a single human labeler as ground truth, but the measured concepts (severity of reality distortion, value judgment distortion) are highly subjective. How reliable is a single human annotator as a gold standard for inherently subjective constructs like disempowerment? Would two or more independent human annotators agree? Even on a small sample (<50), the authors could adopt standard social science practices and report inter-annotator agreement (e.g., Cohen’s kappa) between at least two independent human labelers, and clarify whether the annotator is an author or independently recruited.

7. The analysis depends on Anthropic's Clio framework, which is not open-sourced or widely used. Could the authors provide a more detailed description of how Clio is used in the pipeline in the appendix - specifying where classification prompts are injected and how privacy preservation is concretely implemented? (which seems to be described in https://arxiv.org/abs/2412.13678 but not in this paper) This would be helpful for replication in future research.

**Limitations:**

Yes

**Strengths And Weaknesses:**

Strengths: This work addresses a very relevant and timely problem. If published (and indeed already as a preprint), it is likely to attract significant public and research attention. I want to commend the authors for analyzing real commercial deployment data, which is exceedingly rare and valuable. Turning this into a peer-reviewed contribution with a structured taxonomy is a meaningful service to the field.

Weaknesses: Conceptually, I am afraid the definition of disempowerment is narrowly scoped, limiting the framework's coverage of how AI can reduce human capacity. Operationally, the measurement pipeline can benefit from validity checks and statistical controls.  I elaborate on each point in the Questions below.

---

> ### Author Rebuttal · Authors · 2026-03-30
>
> Thank you for the detailed feedback and suggestions.  To address your key questions in turn:
>
> 1) “... restrict their definition of disempowerment to moral and value-related contexts ...”
>
> Yes, we intentionally filtered out technical domains.  The idea was that execution of technical requests, while sometimes involving consequential decisions or details, usually have fewer opportunities for value expression, and thus were less likely to be a vector for disempowerment.
>
>
> 2) "It would be striking if users face reality distortion 1 out of 7–8 times when asking moral/value-related questions.”
>
> The reviewer raises an important point. The rates reported in the paper use the full 1.5M conversations as their denominator, with screened-out conversations implicitly classified as "none." Since the screener retains only ~7.35% of conversations, rates within this subpopulation are indeed substantially higher. However, we note two important caveats:
>
>  - First, the screener does not specifically filter for “moral/value-relevant conversations”. Instead, it filters for any interaction with potential disempowerment relevance, which includes conversations related to non-moral advice (e.g., someone deciding whether to shower or eat first), mental health questions, etc. We note this only for precision, and it does not change the fact that the finding is striking.
>  - Second, Figure 1 shows disempowerment potential, not actualized disempowerment, which is much more rare. Actualized reality distortion, which is the most common kind of actualized disempowerment we observed, occurred in 0.048% of all interactions, or around 0.65% of screened-in conversations. While any disempowerment is undesirable, the rate of actualized disempowerment is substantially lower than the disempowerment potential the reviewer highlights.
>
>
> 3) “Would the current measurement approach categorize [giving values where the user has none] as disempowerment?”
>
> It would probably be flagged as disempowerment {\emph potential}, at least.  However, we would push back on the idea that someone doesn’t have values on a topic simply by virtue of having not learned, thought about, or discussed it before.  Certainly interlocutors can end up strongly influencing which values people adopt, in some cases to the point where it’s not clear in what sense someone had relevant values beforehand.  But this is a good example of the sort of situation which our classification of “potential” versus “actualized” distortion is meant to address.
>
>
> 4) “... model-level breakdown of disempowerment rates or user approval rates?”
>
> We agree that a model-level breakdown would be valuable. Unfortunately, our analysis pipeline did not retain model information in the classification outputs, so we are unable to provide this breakdown for the current dataset. The reviewer's suggestion that more capable models may produce more persuasive (and therefore potentially more disempowering) responses is worth testing directly in future work, though as we note in the paper, new model releases coincide with shifts in user composition and feedback behavior, making causal attribution to model capabilities alone difficult.
>
>
> 5) “ ... controlling for at least conversation length, domain, and model version?”
>
> We agree that this regression would strengthen the analysis. However, our privacy-preserving classification pipeline produces aggregate and cluster-level outputs rather than row-level records associating individual conversations with their classifications, domain, model version, and feedback signal. Running a regression of the form the reviewer suggests would require retaining these per-conversation associations, which was not possible given the privacy constraints of the current analysis.
>
> In Appendix N, we stated “We considered controlling by domain, but found consistent patterns across all domains”.  We agree that more investigation into potential confounders is needed, however our initial analysis is already quite long.
>
>
> 6) The labeler was one of the authors.  We agree that reporting inter-labeler agreement would be an improvement, but are not planning to gather those results.
>
> 7) “... a more detailed description of how Clio is used in the pipeline ... ?”
>
> We will add a detailed description of how Clio is used in the Appendix. In short, our analysis used the standard Clio pipeline with a custom prompt, where private information is removed through a series of steps (summarization, clustering, and filtering to ensure aggregation minimums are met). The pipeline is fully described by Tamkin et al. (2024).
>
> The prompt modification we made is that schemas are injected into the $\texttt{ \{question\} }$ placeholder in the template described in Appendix G3 and G4 of Tamkin et al. (2024) with $\texttt{< example >}$ removed (as they only apply to open-ended classification). We have added the specific prompt for the schema classifications to the Appendix. All prompt modifications will be provided in a public repo.

---

> > ### Author Rebuttal · Reviewer_gN8F · 2026-04-01
> >
> > Thank you for the detailed clarification, especially on within-screened rates and the Clio pipeline details.
> >
> > My methodological concern regarding inter-rater reliability remains as the authors chose not to pursue this - key constructs like disempowerment can be subjective to annotator perspective, and earlier work on annotating LLM behaviors has indeed reported modest agreement rates across annotators (e.g., https://arxiv.org/pdf/2503.01307). I expect that similar LLM-based measurement approaches will continue to grow in the field, and construct validity of complex LLM behavior measurements (like disempowerment) would be an important question.
> >
> > That said, I understand the practical constraints (and rich qualitative examples the authors provide in the paper partially address the concern). I find the problem important and the dataset valuable - my assessment remains positive.

---

### Official Review · Reviewer_CbH8 · 2026-03-19

**Soundness:** 3
**Presentation:** 3
**Significance:** 3
**Originality:** 4
**Overall Recommendation:** 4
**Confidence:** 4

**Summary:**

This paper presents a large-scale empirical study of "situational disempowerment" in real-world LLM interactions, using 1.5 million Claude.ai conversations analyzed via a privacy-preserving pipeline. The authors introduce a conceptual framework distinguishing three primitives of disempowerment potential (reality distortion, value judgment distortion, and action distortion) and identify several "amplifying factors" such as authority projection and user vulnerability.
Overall, the article's main finding concerns the empirical observation that severe disempowerment potential is relatively rare in percentage terms but non-negligible at scale, and that these concerning interactions are often positively rated by users.
This work considers a critical issue regarding how widespread LLM usage may subtly undermine human autonomy, even when interactions appear helpful or satisfying.

**Compliance With Llm Reviewing Policy:**

Affirmed.

**Key Questions For Authors:**

1. What are the key implications or takeaways that you want readers to have? And who is the target audience?
2. Are there ways to address those takeaways through governance or policy? Or should the follow on work focus on technical approaches?
3. Are the framework and primitives derived from existing literature? If so, what? If not, how did you decide on these characteristics specifically?

**Limitations:**

The authors sufficiently cover technical limitations. But the significance of the work hinges on the societal impacts. As mentioned above, the paper would greatly benefit from further discussion on the implications.

**Strengths And Weaknesses:**

# Soundness
## Strengths:
The paper is generally methodologically sound. The use of a large-scale dataset (1.5M real-world conversations) provides strong empirical grounding, and the privacy-preserving analysis pipeline is appropriate for the sensitive nature of the data. The combination of LLM-based classification with validation against human labels (showing high agreement) increases confidence in the measurement approach. The mixed-methods design which combines quantitative prevalence estimates with qualitative clustering adds depth and triangulation to the findings.
## Weaknesses:
A key limitation is the reliance on "disempowerment potential" rather than directly observed outcomes. While the authors appropriately acknowledge this constraint, the interpretation of results sometimes leans toward stronger claims than the evidence supports. My main example is where the authors suggest that for some aspects, they assert that no (or little) evidence of actualized distortion does not disprove actualized distortion, but this comes across as undermining the existing findings. This makes it difficult to assess the real-world severity of the identified phenomena. Additionally, the classification and clustering pipeline, while validated, may still introduce systematic biases or abstraction artifacts that are not fully investigated. This was also mentioned in the paper as a potential limitation, but a solution or mitigation was not addressed.
# Presentation
## Strengths:
The paper is clearly written and well-structured overall. The conceptual framework is introduced in an intuitive way, and the organization from framework through qualitative analysis is logical and easy to follow. The qualitative examples and cluster summaries are especially effective at making the abstract risks concrete. The figures are informative and help illustrate key patterns.
## Weaknesses:
The main presentation gap is the absence of a dedicated discussion section. While results are clearly presented, the paper does not sufficiently synthesize or interpret them. In particular, the reader is left without clear guidance on questions like
- How these findings translate into real-world harms
- Types of users who may be at higher risk (diagnosed/undiagnosed mental illnesses etc)
- What the implications mean for system design, deployment, or policy
- How practitioners should respond (mitigation strategies, product decisions, etc)
- Whether certain risks are acceptable vs. unacceptable
- How this framework should influence or inform future evaluation or policy

Given the strong normative framing around "empowerment," this lack of interpretive synthesis limits the impact of the narrative. Additionally, the conceptual framing of "disempowerment" would benefit from stronger grounding in existing literature to clarify how the framework/definition and the proposed primitives (reality, values, actions) were derived.
# Significance
## Strengths:
The paper addresses a highly important and interesting problem, focusing on how LLM interactions may influence human autonomy and decision-making. The scale and ecological validity of the dataset make the findings particularly valuable, as they reflect real-world usage rather than controlled settings. The empirical results, especially the concentration of risk in personal domains and the positive user ratings for potentially disempowering interactions, have meaningful implications for AI alignment, product design, and evaluation practices. The work has strong potential to influence future research on human-AI interaction and safety.
## Weaknesses:
As stated above, the significance is somewhat limited by the lack of a clear articulation of a "so what." Without a deeper discussion of implications, it is difficult to translate the findings into actionable insights for researchers or practitioners. For example, the paper does not clearly address whether the observed risks warrant intervention, how severe they are in practice, or what tradeoffs (for example, helpfulness vs. autonomy) should be considered. As a result, the contribution remains mainly descriptive rather than prescriptive.
# Originality
## Strengths:
The paper offers a novel and compelling framing of "situational disempowerment" that bridges multiple areas, including alignment, human-AI interaction, and sociotechnical risk. The decomposition into reality, value judgement, and action distortion provides a useful lens for analyzing LLM impacts. The application of this framework to large-scale real-world data is also original and represents a meaningful contribution. Additionally, the integration of privacy-preserving analytics with behavioral classification is a creative and practically relevant methodological combination.
## Weaknesses:
While the framing is novel, it is not always clearly situated within existing theoretical or philosophical literature. The paper would benefit from more explicit connections to some prior work on autonomy, agency, and decision-making to better justify the choice of primitives and clarify what is substantively new versus adapted. Strengthening this grounding would make the originality more legible and robust.

---

> ### Author Rebuttal · Authors · 2026-03-30
>
> Thank you for the detailed feedback and suggestions.  We mostly agree with your analysis of the shortcomings of the method and paper.  However, some of them we feel are difficult to address without making normative claims.  For instance, we’re not sure how to “address whether the observed risks warrant intervention”, though this is certainly an important question.
>
> We do cite and make some explicit connections to some prior work on autonomy, agency, and decision-making, but agree that more such connections would improve the paper, and we will include a few in the final draft, such as:
>
> 1. Laitinen & Pessi, "AI Systems and Respect for Human Autonomy" (Frontiers in AI, 2021)
> https://www.frontiersin.org/journals/artificial-intelligence/articles/10.3389/frai.2021.705164/full
> 2. Susser, Roessler & Nissenbaum, "Online Manipulation: Hidden Influences in a Digital World" (Georgetown Law Technology Review, 2019)
> https://papers.ssrn.com/sol3/papers.cfm?abstract_id=3306006
> 3. Carmona-Díaz, Jiménez-Leal, Vold et al., "Autonomy by Design: Preserving Human Autonomy in AI Decision-Support" (Philosophy & Technology, 2025)
> https://link.springer.com/article/10.1007/s13347-025-00932-2
> 4. Calvo, Peters, Vold & Ryan, "Supporting Human Autonomy in AI Systems: A Framework for Ethical Enquiry" (Springer, 2020)
> https://link.springer.com/chapter/10.1007/978-3-030-50585-1_2
> 5. Christman, "Autonomy in Moral and Political Philosophy" (Stanford Encyclopedia of Philosophy)
> https://plato.stanford.edu/entries/autonomy-moral/
>
> We’re all ears for relevant work we might have missed.
>
>
> To address your key questions in turn:
>
> 1) What are the key implications or takeaways that you want readers to have? And who is the target audience?
>
> Since this is an exploratory empirical work, our first priority is to give as clear a window into a broad set of phenomena that we think are important and under-investigated.  We hope to provide enough results that a careful or more knowledgeable reader can find insights that we miss.  That being said, perhaps the main takeaway is simply that disempowering-seeming interactions and relationships seem to be a relatively small but consistent feature of contemporary LLM usage.  And it raises and provides context for two problems for LLM developers: First, the technical problem of avoiding some of these behaviors which are presumably unwanted by all parties, such as endorsing user delusions.  Second, the policy problem of addressing user-incentivized behaviors that nevertheless might not be in their best long-term interests.
>
> 2) Are there ways to address those takeaways through governance or policy? Or should the follow on work focus on technical approaches?
>
> As above, we think that the incentive question should be addressed by governance and policy, while the unwanted behavior problem should be addressed by technical approaches (similar to avoiding jailbreaking).
>
>
> 3) Are the framework and primitives derived from existing literature? If so, what? If not, how did you decide on these characteristics specifically?
>
> The main source of inspiration for this analysis was:
> Human Autonomy at Risk? An Analysis of the Challenges from AI by C. Prunkl
> https://link.springer.com/article/10.1007/s11023-024-09665-1
> However, the framework was derived mainly from our own first-principles thinking.
>
> The primitives were derived from ad-hoc exploratory analysis of thumbs data.

---

> > ### Author Rebuttal · Reviewer_CbH8 · 2026-04-03
> >
> > I appreciate the clarifications provided, particularly around the intended scope of the paper as exploratory empirical work and the spotlight on two main downstream problems. This helps clarify the intended contribution and audience.
> >
> > The rebuttal partially addresses my concerns regarding the lack of a clear “so what,” especially by outlining high-level implications for both technical and governance directions. I also appreciate the additional references provided to better ground the conceptual framing in existing literature on autonomy.
> >
> > However, a few concerns remain only partially resolved:
> > - Discussion / implications: My concern is primarily about the lack of a dedicated discussion section that synthesizes implications, clarifies the scope of claims, and guides readers on considerations about the findings. Addressing this in the final version would substantially strengthen the paper.
> > - Normative framing: I understand the authors’ hesitation to make strong normative claims. However, given that the paper is explicitly framed around "disempowerment," some level of interpretive guidance seems warranted. A scoped or conditional discussion of implications (even with 'considerations' rather than 'normative requirements') would improve clarity.
> > - Framework grounding: The additional citations and explanation are helpful. However, the fact that the primitives are partly derived from ad-hoc analysis doesn't come across strongly, and further clarification in the paper would be valuable to strengthen the theoretical grounding and make the contribution more clear.
> >
> > Overall, I find the rebuttal helpful and clarifying, but I still believe that addressing these points in the paper would be important for maximizing the paper’s impact.
> >
> > Follow-up Questions for Authors:
> > - Do the authors plan to include a dedicated discussion section in the final version that explores/considers the key implications?
> > - Can the authors more explicitly incorporate the cited autonomy literature into the framing of the primitives to strengthen the theoretical grounding, and explicitly mention and describe the ad hoc methodology for coming up with the framework?

---

### Decision · Program_Chairs · 2026-04-30

**Decision:**

Accept (regular)

**Comment:**

Reviewers agreed that the paper tackles an important problem (i.e., disempowerment patterns in real-world AI assistant interactions) and offers a solid mixed-method approach to evaluation. They also appreciated that the evaluations were conducted on a large-scale real-world dataset. But they questioned key elements of the framework offered, including the definition of disempowerment, the validity of the evaluation pipeline, and the strength of the link between the experimental results and interpretations offered. They also raised concerns about the paper's framing and its grounding in the relevant literature, and asked for a detailed discussion section addressing the real-world implications of the work. The authors acknowledged and addressed some of the issues raised, including grounding in the relevant literature. Considering that the reviewers are overall positive on the paper and assuming that the authors will clearly acknowledge and discuss the limitations and implications of their work, I recommend acceptance.